# An Interval Two-Stage Stochastic Programming Model for Flood Resources Allocation under Ecological Benefits as a Constraint Combined with Ecological Compensation Concept

**DOI:** 10.3390/ijerph16061033

**Published:** 2019-03-21

**Authors:** Yu Qiu, Yuan Liu, Yang Liu, Yingzi Chen, Yu Li

**Affiliations:** 1Northeast Asian Studies College, Jilin University, Changchun 130012, China; qiuyu075@163.com; 2Key Laboratory of Environmental Change and Natural Disaster, MOE, Beijing Normal University, Beijing 100875, China; ryanliu_xy@163.com; 3School of Water Resources and Environment Engineering, Changchun Institute of Technology, Changchun 130012, China; liuyangbktzzy@163.com; 4College of Environmental Science and Engineering, North China Electrical Power University, Beijing 102206, China

**Keywords:** wetland, flood resources allocation, interval two-stage stochastic programming, ecological compensation, ecological benefits

## Abstract

The Momoge National Nature Reserve (MNNR) is located at the intersection of Nenjiang and Taoer Rivers in Baicheng City, Jilin Province, where the Taoer River is the main source of water for the nature reserve. However, due to the construction of the water control project in the upper reaches of the Taoer River, the MNNR has been in a state of water shortage for a long time. To guarantee the wetland function of the nature reserve, the government planned to carry out normal and flood water supply from Nenjiang River through the West Water Supply Project of Jilin Province. Therefore, how to improve the utilization of flood resources effectively has become one of the key issues of ecological compensation for the MNNR. In this paper, a flood resources optimal allocation model that is based on the interval two-stage stochastic programming method was constructed, and the corresponding flood resource availability in different flow scenarios of Nenjiang River were included in the total water resources to improve their utilization. The results showed that the proportion of flood resources that were used in the MNNR after optimization was more than 70% under different flow scenarios, among which the proportion of flood resources under a low-flow scenario reached 77%, which was 23% higher than the proposed increase. In addition, the ecological benefits of low, medium, and high flow levels reached the range of 26.30 (10^6^ CNY) to 32.14(10^6^ CNY), 28.21(10^6^ CNY) to 34.49(10^6^ CNY) and 29.41(10^6^ CNY) to 35.94(10^6^ CNY), respectively. According to the results, flood resources significantly reduce the utilization of normal water resources, which can be an effective supplement to the ecological compensation of nature reserves and provide a basis for the distribution of transit flood resources in other regions.

## 1. Introduction

The shortage of water resources has become a main contradiction of water problems in China, especially in the northern region. On the one hand, water resources are generally scarce. On the other hand, to ensure the safety of rivers and dams during flood seasons, the government has to devote considerable effort to channeling floods into the sea, which results in the loss of a large amount of valuable water resources that have not been fully utilized [1]. The government work document in 2011 clearly pointed out that it is necessary to significantly improve the utilization of rainwater resources and water supply, to continue to promote the ecological restoration of ecologically fragile rivers and areas, and to establish and improve the ecological compensation system [2]. The use of flood resources is a process that takes engineering measures to accumulate floods or rainwater as a source of available water in flood seasons. It can not only alleviate the shortage of water resources in the region and improve the ecological environment, but can also effectively reduce the pressure on flood discharge facilities and reduce flood damage [3,4,5,6].

The use of flood resources in the basin is an important measure to exploit the utilization potential of flood resources, alleviate water shortages, reduce losses from floods and droughts, and achieve sustainable development and utilization of water resources, which can be regarded as a specific form of ecological compensation [7,8,9]. Xiang et al. [10] expounded the domestic and foreign history of flood resource utilization. Tao et al. [11] analyzed the problems of flood resource utilization in China. Some researchers have discussed the implementation methods of flood resource utilization [12]. In the Panjiakou Reservoir of the Weihe River Basin, the cluster analysis method was used to carry out quantitative flood season staging research, which provides an important prerequisite for the development of flood resources and the adjustment of the flood season limit [13]. Luo et al. [14] constructed a potential model for the development of flood resources and calculated and analyzed the potential of flood resource utilization in the lower reaches of the basin. However, most of the current research on the utilization of flood resources is based on a feasibility analysis that was conducted at the theoretical level, while there are few studies on the utilization and optimization of flood resources that are based on practice.

Studying the allocation of flood resources in various fields, such as production, living, and the ecological environment, can further strengthen and enhance our understanding of flood resources in various sectors and provide an important theoretical basis for the development of flood resources [15]. At present, research on water resources allocation methods is becoming more comprehensive, but research on flood resource allocation is still in the development stage. Afzal et al. [16] constructed a regional irrigation system linear programming model to optimize the allocation of water resources of different water qualities. Dudley [17] proposed a coupled model of the crop growth model with stochastic dynamic programming and optimized the seasonal irrigation water consumption in irrigation districts. Kumar et al. [18] constructed an urban sewage discharge optimization model and proposed a regional water quality management plan that meets both technical and economic requirements. Lu et al. [19] coupled fuzzy programming with the stochastic programming method to form an interval fuzzy two-stage stochastic programming model for water resources optimization management. Cardwell et al. [20] studied a stochastic dynamic programming model for the optimal allocation and management of pollution loads in multiple point source regions. Xie et al. [21] integrated interval linear programming and stochastic chance-constrained programming to form a non-deterministic chance-constrained stochastic water quality management planning model, which was applied to wastewater discharge management in coastal areas. Meng et al. [22] developed a two-stage stochastic programming (TSP) model to support the regional waste load (chemical oxygen demand (COD) and NH_3-_N) allocation in four main pollution departments (industry, municipal, livestock breeding, and agriculture). Fu et al. [23] considered the uncertainty of the optimal allocation of water resources in irrigation districts and established a two-stage stochastic programming method, which was applied in three irrigation districts in China. Wang et al. [24] combined the analytic hierarchy process with the linear programming method to construct an optimal water resource allocation model for the water receiving area of the South-to-North Water Transfer Project.

In view of the lack of research on the optimal allocation of flood resources, we constructed an optimal allocation model of flood resource based on the stochastic programming method in this paper. Simultaneously, the model aims at maximizing ecological benefits. This study takes the utilization of flood resources in the Momoge National Nature Reserve (MNNR) in Jilin Province as an example to solve the problem of using ecological resources for ecological compensation.

## 2. Case Descriptions

### 2.1. Project Overview

The West Water Supply Project of Jilin Province is located in the western plain area on the right bank of Nenjiang River and on the left bank of Songhua River and covers 8 counties. The study area is located in an internationally important wetland and mainly relies on the “Yinnenrubai” water supply project, which connects 58 lakes around the 52 km main canal and its associated water channels. The West Water Supply Project of Jilin Province is an ecological water transfer project, which is both a water conservation project and an ecological project. According to “Planning report on comprehensive utilization of rivers and lakes for rainwater and flood resources in western Jilin Province” (hereinafter referred to as the “Planning Report”) [25], water diversion is given priority to ensure the water needs of the MNNR. Therefore, 15 lakes in the MNNR are selected for this study. The water diversion node and water diversion process are shown in Figure 1. The water diversion period of Nenjiang River is from July to October. The three water intakes for the MNNR are the third branch channel of the Baishatan irrigation area (TBC), the Shijianfang intake gate (SIG) and the Haernao pumping station (HPS). The annual water intake is approximately 11.715 million m^3^.

### 2.2. Ecological Benefits Evaluation Index System

The ecosystem service function refers to the natural environmental conditions upon which human beings depend for the formation and maintenance of ecosystems and ecological processes. Life support products and services are directly or indirectly obtained through the structure, process, and function of ecosystems [26,27,28]. Ecosystem service functions can be summarized into three categories. The first category is the production of substances in the ecosystem, including the functions that the wetland ecosystem can provide for human beings, such as food, industrial raw materials, etc. The second category is the environmental regulation of ecosystems, including flood regulation, microclimate regulation, carbon sequestration, atmospheric regulation, plant adsorption, and biodiversity [29]. The third category is the service functions for the humanities, including the value of scientific research, tourism resources, and landscape functions. Ecosystem service functions include not only the ability of ecosystems to directly supply products or raw materials for human production and consumption but also the ability of ecological environment elements to improve the quality of life, while maintaining the environmental conditions and effects that human beings depend on for survival [30]. Therefore, it is important to establish an ecosystem service function evaluation indicator system that is suitable for this study.

The study combines the benefits of water conservancy projects, ecological benefits, and economic benefits. According to the “Planning Report” and regional environmental characteristics, we have constructed 11 service function evaluation index systems, as shown in Table 1. The evaluation method of the indicator layer is shown in Appendix A.

## 3. Interval Two-Stage Flood Resources Allocation Model

The optimal allocation of water resources involves meteorological conditions, the ecological functions of structure, and other factors, which is a complex system. Different water supply methods are affected by the amount of incoming water, which has strong randomness. This requires that the relevant water supply project must make timely decisions at different probability levels of random variables at each stage. It can be solved by a stochastic programming model, which can reflect the unpredictability of the system in water resources management. When the realization of random variables in the future stage is unpredictable, it is necessary to make decisions on the current stage [31].

Managers are required to make advanced decisions on the amount of water in the current year and determine the initial water supply target for different ecological functions. Since the water supply situation of different water supply projects is random, once the water supply quantity does not reach the initial water supply target, it is necessary to reduce the water supply or externally adjust the water source for water supply, thereby causing additional costs and the corresponding consequences. Therefore, to maximize the ecological benefits of the MNNR, this paper introduces the water shortage penalty coefficient according to the different impacts of various ecological functions during water shortage on the final ecological benefits to determine the water demand degree of different ecological functions, as a means implementing procedures that will minimize the economic penalty to its lowest degree.

### 3.1. Model Formulation

In this paper, the total water cost and the water shortage penalty coefficient are introduced under the constraints of water availability, the ecological functional area, and the water replenishment sequence. Determining the water cost under the condition of sufficient water replenishment in the lakes of the nature reserve is the first stage. Determining the ecological benefits loss when the water supply quantity fails to meet the requirements is the second stage, for which a two-stage stochastic programming model is established. Since the water resources system is a huge complex system that contains many uncertain factors, it is difficult for policy makers to predict the water supply of the water source. Therefore, the initial water supply target W is uncertain, the difference between the water source and the water supply route leads to the uncertainty of the water supply cost T, and changes in the value of ecological functions and water consumption make the water shortage penalty coefficient C uncertain. To express this uncertainty, this study introduces interval parameters, with “+” for the upper limit of the parameter, “−“ for the lower limit of the parameter, and a two-stage stochastic programming model for the interval:(1)$$Maxf^{\pm }=\overset{15}{\underset{i=1}{\sum }}\overset{4}{\underset{j=1}{\sum }}A_{ij}^{\pm }\cdot Y_{ij}\cdot B_{j}-\overset{2}{\underset{m=1}{\sum }}\overset{3}{\underset{n=1}{\sum }}T_{mn}^{\pm }\cdot W_{mn}^{\pm }-E(\overset{2}{\underset{m=1}{\sum }}\overset{3}{\underset{n=1}{\sum }}C_{mn}^{\pm }S_{mn}^{\pm })$$
where *S* is the water shortage in the forecast year, 10^4^ m^3^. This represents the decision variable of the second stage, which is affected by the water quantity and the rainfall in the current year and is thus difficult to determine. Therefore, the water shortage under different forecasting water quantities in the forecast year is treated as a discrete function, and the probability of occurrence of different water levels is assumed to be *P_h_*, where *h* = 1, 2, 3. When *h* = 1, it means that there is enough water in the forecast year, which is considered to be high flow, and the water shortage is the smallest; when *h* = 2, it means that the water flow is considered to be moderate, and the water shortage is small; and when *h* = *3*, it means the water flow is considered to be lowest, and the water shortage is large. In addition, *P*_h_
*= P_1_, P_2_, P_3_*, $\sum_{h=1}^{3}P_{h}=1$. *E* represents a mathematical expectation function. Therefore, the interval two-stage stochastic programming model can be expressed in the following form:(2)$$Maxf^{\pm }=\sum_{i=1}^{15}\sum_{j=1}^{4}A_{ij}^{\pm }\cdot Y_{ij}\cdot B_{j}^{\pm }-\sum_{m=1}^{2}\sum_{n=1}^{3}T_{mn}^{\pm }\cdot W_{mn}^{\pm }-\sum_{m=1}^{2}\sum_{n=1}^{3}C_{mn}^{\pm }(\sum_{h=1}^{3}P_{h}\cdot S_{mnh}^{\pm })$$
where *f* is the economic benefits, yuan; *A* is the lake area, m^2^; *Y* is 0–1 variable; *B* is the unit ecological function service value, yuan per m^2^; *T* is the water supply cost, yuan/m^3^; *W* is the forecasted water volume for the target year, m^3^; *C* is the water shortage penalty coefficient, that is, the loss when the actual water supply fails to reach the predicted value, yuan/m^3^; *P* is the probability of occurrence of different flow levels in the forecast year; *i* indicates 15 lakes in the MNNR; *j* represents different ecological functional areas, wherein *j* = 1, 2, 3, 4 represent fish farming, crab raising, reed wetland, and marsh wetland, respectively; *m* represents different water resources; *m* = 1, 2 represents normal water and flood resources; *n* represents 3 water intakes in the study area; and *n* = 1, 2, 3 represent the third branch channel of Baishatan irrigation area (TBC), Shijianfang intake gate (SIG), and the Haernao pumping station (HPS), respectively.


**Constraint condition:**


1. Minimum water constraint
(3)$$\sum_{m=1}^{2}\sum_{n=1}^{1}W_{mn}^{\pm }\geq \sum_{i=1}^{9}\sum_{j=1}^{4}PA_{ij}^{\pm }\cdot U_{ij}^{\pm }$$
(4)$$\sum_{m=1}^{2}\sum_{n=2}^{2}W_{mn}^{\pm }\geq \sum_{i=10}^{13}\sum_{j=1}^{4}PA_{ij}^{\pm }\cdot U_{ij}^{\pm }$$
(5)$$\sum_{m=1}^{2}\sum_{n=3}^{3}W_{mn}^{\pm }\geq \sum_{i=14}^{15}\sum_{j=1}^{4}PA_{ij}^{\pm }\cdot U_{ij}^{\pm }$$
where $PA_{ij}^{\pm }$ is the planned area of different ecological functions of each lake, m^2^; and *U_j_* is the water demand per unit area of different ecological functions, m^3^/m^2^. In addition, the three water intakes correspond to three groups of lakes (Figure 1). Therefore, the minimum water amount constraints are imposed on the three water intakes.

2. Water replenishment ability constraint
(6)$$\sum_{m=1}^{2}\sum_{n=1}^{3}(W_{mn}^{\pm }-S_{mn}^{\pm })\leq \sum_{n=1}^{3}Q_{n}^{\pm },\forall m,n$$

*Q_ik_* is the amount of water that can be replenished by different water resources in the lakes of the MNNR during the early stage of planning, regardless of the loss of water delivery.

3. Area constraint
(7)$$\sum_{j=1}^{4}A_{ij}^{\pm }\cdot Y_{ij}\leq A_{i\max}^{\pm },\forall i$$

The area of the lakes cannot be expanded indefinitely. After optimization, the sum of the area of each ecological function area of each lake should be less than or equal to the original area of the lake, and $A_{i\max}^{\pm }$ is the largest area of the *i* lake bubble.

4. Water supply order constraint

The water supply is carried out from the inside out in the order of the lake, the reed wetland, and the swamp wetland; that is, the fish culture and the crab-raising ecological function of the lake are preferentially satisfied, then the reed wetland is supplied with water, and, finally, the marsh wetland is supplied with water.

As for TBC, when $\sum_{i=1}^{9}\sum_{j=1}^{2}A_{ij}^{\pm }\cdot U_{ij}^{\pm }\geq \sum_{i=1}^{9}\sum_{j=1}^{2}A_{ij\min}\cdot U_{ij}^{\pm }$,
(8)$$\sum_{i=1}^{9}\sum_{j=3}^{4}A_{ij}^{\pm }\cdot U_{ij}^{\pm }=\sum_{n=1}^{1}Q_{n}^{\pm }-\sum_{i=1}^{9}\sum_{j=1}^{2}A_{ij}^{\pm }\cdot U_{ij}^{\pm }$$
when $\sum_{i=1}^{9}\sum_{j=1}^{2}A_{ij}^{\pm }\cdot U_{ij}^{\pm }\leq \sum_{i=1}^{9}\sum_{j=1}^{2}A_{ij\min}\cdot U_{ij}^{\pm }$,
(9)$$\sum_{i=1}^{9}\sum_{j=3}^{4}A_{ij}^{\pm }\cdot U_{ij}^{\pm }=0$$

As for SIG, when $\sum_{i=10}^{13}\sum_{j=1}^{2}A_{ij}^{\pm }\cdot U_{ij}^{\pm }\geq \sum_{i=10}^{13}\sum_{j=1}^{2}A_{ij\min}\cdot U_{ij}^{\pm }$,
(10)$$\sum_{i=10}^{13}\sum_{j=3}^{4}A_{ij}^{\pm }\cdot U_{ij}^{\pm }=\sum_{n=2}^{2}Q_{n}^{\pm }-\sum_{i=10}^{13}\sum_{j=1}^{2}A_{ij}^{\pm }\cdot U_{ij}^{\pm }$$
when $\sum_{i=10}^{13}\sum_{j=1}^{2}A_{ij}^{\pm }\cdot U_{ij}^{\pm }\leq \sum_{i=10}^{13}\sum_{j=1}^{2}A_{ij\min}\cdot U_{ij}^{\pm }$,
(11)$$\sum_{i=10}^{13}\sum_{j=3}^{4}A_{ij}^{\pm }\cdot U_{ij}^{\pm }=0$$

As for HPS, when $\sum_{i=14}^{15}\sum_{j=1}^{2}A_{ij}^{\pm }\cdot U_{ij}^{\pm }\geq \sum_{i=14}^{15}\sum_{j=1}^{2}A_{ij\min}\cdot U_{ij}^{\pm }$,
(12)$$\sum_{i=14}^{15}\sum_{j=3}^{4}A_{ij}^{\pm }\cdot U_{ij}^{\pm }=\sum_{n=3}^{3}Q_{n}^{\pm }-\sum_{i=14}^{15}\sum_{j=1}^{2}A_{ij}^{\pm }\cdot U_{ij}^{\pm }$$
when $\sum_{i=14}^{15}\sum_{j=1}^{2}A_{ij}^{\pm }\cdot U_{ij}^{\pm }\leq \sum_{i=14}^{15}\sum_{j=1}^{2}A_{ij\min}\cdot U_{ij}^{\pm }$,
(13)$$\sum_{i=14}^{15}\sum_{j=3}^{4}A_{ij}^{\pm }\cdot U_{ij}^{\pm }=0$$

5. Non-negative constraint
(14)$$W_{mn}^{\pm }\geq S_{mn}^{\pm }\geq 0,\forall m,n$$
(15)$$A_{ij}^{\pm }\geq 0$$

### 3.2. Model Solution

According to the characteristics of the model itself, $W_{mn}^{\pm }$ is an uncertain number that is expressed in interval form. It is difficult to judge the value of the system when it is worth the minimum. Therefore, the solution method provided in Reference [31] introduces another decision variable $z_{mn},z_{mn}\in [0,1]$. Let $W_{mn}^{\pm }=W_{mn}^{-}+\Delta W_{mn}z_{mn}$, when $z_{mn}=1$, $W_{mn}^{\pm }$ takes the upper limit value, when $z_{mn}=0$, $W_{mn}^{\pm }$ takes the lower limit value and $\Delta W_{mn}=W_{mn}^{+}-W_{mn}^{-}$ is a certain value. The optimal value $z_{mnopt}$ can be obtained by introducing the decision variable $z_{mn}$, thereby obtaining the optimal value $W_{mn}^{\pm }=W_{mn}^{-}+\Delta W_{mn}z_{mnopt}$ of $W_{mn}^{\pm }$. Therefore, the water supply value of the ecological function value can be determined, the upper limit value of the model can be solved to obtain $f_{opt}^{+}$ and $S_{mnopt}^{+}$, and the optimal water resource allocation scheme can finally be determined.

The above model is transformed into two sub-models to solve. First, the upper bound sub-model should be solved, which can be expressed as:(16)$$\begin{array}{c} Maxf^{+}=\sum_{i=1}^{15}\sum_{j=1}^{4}A_{ij}^{+}\cdot Y_{ij}\cdot B_{j}^{+}-\sum_{m=1}^{2}\sum_{n=1}^{3}T_{mn}^{-}\cdot (W_{mn}^{-}+\Delta W_{mn}z_{mn})\\ -\sum_{m=1}^{2}\sum_{n=1}^{3}C_{mn}^{-}(\sum_{h=1}^{3}P_{h}\cdot S_{mnh}^{-}) \end{array}$$

Constraint condition:(17)$$\sum_{m=1}^{2}\sum_{n=1}^{3}\left(W_{mn}^{-}+\Delta W_{mn}z_{mn}\right)\geq \sum_{i=1}^{15}\sum_{j=1}^{4}PA_{ij}\cdot U_{ij}^{-},\forall i,j,m,n$$
(18)$$\sum_{m=1}^{2}\sum_{n=1}^{3}(W_{mn}^{-}+\Delta W_{mn}z_{mn})-S_{mn}^{-}\leq \sum_{n=1}^{3}Q_{n}^{-},\forall m,n$$
(19)$$\sum_{j}^{4}A_{ij}^{+}\cdot Y_{ij}\leq A_{i\max}^{+},\forall i$$

As for TBC, when $\sum_{i=1}^{9}\sum_{j=1}^{2}A_{ij}^{+}\cdot U_{ij}^{-}\geq \sum_{i=1}^{9}\sum_{j=1}^{2}A_{ij\min}\cdot U_{ij}^{-}$,
(20)$$\sum_{i=1}^{9}\sum_{j=3}^{4}A_{ij}^{+}\cdot U_{ij}^{-}=\sum_{n=1}^{1}Q_{n}^{-}-\sum_{i=1}^{9}\sum_{j=1}^{2}A_{ij}^{+}\cdot U_{ij}^{-}$$
when $\sum_{i=1}^{9}\sum_{j=1}^{2}A_{ij}^{+}\cdot U_{ij}^{-}\leq \sum_{i=1}^{9}\sum_{j=1}^{2}A_{ij\min}\cdot U_{ij}^{-}$,
(21)$$\sum_{i=1}^{9}\sum_{j=3}^{4}A_{ij}^{+}\cdot U_{ij}^{-}=0$$

As for SIG, $\sum_{i=10}^{13}\sum_{j=1}^{2}A_{ij}^{+}\cdot U_{ij}^{-}\geq \sum_{i=10}^{13}\sum_{j=1}^{2}A_{ij\min}\cdot U_{ij}^{-}$,
(22)$$\sum_{i=10}^{13}\sum_{j=3}^{4}A_{ij}^{+}\cdot U_{ij}^{-}=\sum_{n=2}^{2}Q_{n}^{-}-\sum_{i=10}^{13}\sum_{j=1}^{2}A_{ij}^{+}\cdot U_{ij}^{-}$$
when, $\sum_{i=10}^{13}\sum_{j=1}^{2}A_{ij}^{+}\cdot U_{ij}^{-}\leq \sum_{i=10}^{13}\sum_{j=1}^{2}A_{ij\min}\cdot U_{ij}^{-}$,
(23)$$\sum_{i=10}^{13}\sum_{j=3}^{4}A_{ij}^{+}\cdot U_{ij}^{-}=0$$

As for HPS, $\sum_{i=14}^{15}\sum_{j=1}^{2}A_{ij}^{+}\cdot U_{ij}^{-}\geq \sum_{i=14}^{15}\sum_{j=1}^{2}A_{ij\min}\cdot U_{ij}^{-}$,
(24)$$\sum_{i=14}^{15}\sum_{j=3}^{4}A_{ij}^{+}\cdot U_{ij}^{-}=\sum_{n=3}^{3}Q_{n}^{-}-\sum_{i=14}^{15}\sum_{j=1}^{2}A_{ij}^{+}\cdot U_{ij}^{-}$$
when, $\sum_{i=14}^{15}\sum_{j=1}^{2}A_{ij}^{+}\cdot U_{ij}^{-}\leq \sum_{i=14}^{15}\sum_{j=1}^{2}A_{ij\min}\cdot U_{ij}^{-}$,
(25)$$\sum_{i=14}^{15}\sum_{j=3}^{4}A_{ij}^{+}\cdot U_{ij}^{-}=0$$
(26)$$(W_{mn}^{-}+\Delta W_{mn}z_{mn})\geq S_{mn}^{-}\geq 0,\forall m,n$$
(27)$$0\leq z_{mn}\leq 1,\forall m,n$$

For this model, $S_{mn}^{-}$ and $z_{mn}$ are the decision variables, $A_{ijopt}^{+}$, $S_{mnopt}^{-}$, $z_{mnopt}$, and $f_{opt}^{+}$ are solutions to the model, and the best water supply target is $W_{mnopt}^{\pm }=W_{mn}^{-}+\Delta W_{mn}z_{mnopt}$. Based on the interactive algorithm, the sub-model that meets the lower limit of the objective function is as follows:(28)$$\begin{array}{c} Maxf^{-}=\sum_{i=1}^{15}\sum_{j=1}^{4}A_{ij}^{-}\cdot Y_{ij}\cdot B_{j}^{-}-\sum_{m=1}^{2}\sum_{n=1}^{3}T_{mn}^{+}\cdot (W_{mn}^{-}+\Delta W_{mn}z_{mnopt})\\ -\sum_{m=1}^{2}\sum_{n=1}^{3}C_{mn}^{+}(\sum_{h=1}^{3}P_{h}\cdot S_{mnh}^{+}) \end{array}$$

Constraint condition:(29)$$\sum_{m=1}^{2}\sum_{n=1}^{3}(W_{mn}^{-}+\Delta W_{mn}z_{mn})-S_{mn}^{+}\leq \sum_{n=1}^{3}Q_{n}^{+},\forall m,n$$
(30)$$\sum_{j}^{4}A_{ij}^{-}\cdot Y_{ij}\leq A_{i\max}^{-},\forall i$$

As for TBC, when $\sum_{i=1}^{9}\sum_{j=1}^{2}A_{ij}^{-}\cdot U_{ij}^{+}\geq \sum_{i=1}^{9}\sum_{j=1}^{2}A_{ij\min}\cdot U_{ij}^{+}$,
(31)$$\sum_{i=1}^{9}\sum_{j=3}^{4}A_{ij}^{-}\cdot U_{ij}^{+}=\sum_{n=1}^{1}Q_{n}^{+}-\sum_{i=1}^{9}\sum_{j=1}^{2}A_{ij}^{-}\cdot U_{ij}^{+}$$
when, $\sum_{i=1}^{9}\sum_{j=1}^{2}A_{ij}^{-}\cdot U_{ij}^{+}\leq \sum_{i=1}^{9}\sum_{j=1}^{2}A_{ij\min}\cdot U_{ij}^{+}$,
(32)$$\sum_{i=1}^{9}\sum_{j=3}^{4}A_{ij}^{-}\cdot U_{ij}^{+}=0$$

As for SIG, $\sum_{i=10}^{13}\sum_{j=1}^{2}A_{ij}^{-}\cdot U_{ij}^{+}\geq \sum_{i=10}^{13}\sum_{j=1}^{2}A_{ij\min}\cdot U_{ij}^{+}$,
(33)$$\sum_{i=10}^{13}\sum_{j=3}^{4}A_{ij}^{-}\cdot U_{ij}^{+}=\sum_{n=2}^{2}Q_{n}^{+}-\sum_{i=10}^{13}\sum_{j=1}^{2}A_{ij}^{-}\cdot U_{ij}^{+}$$
when, $\sum_{i=10}^{13}\sum_{j=1}^{2}A_{ij}^{-}\cdot U_{ij}^{+}\leq \sum_{i=10}^{13}\sum_{j=1}^{2}A_{ij\min}\cdot U_{ij}^{+}$,
(34)$$\sum_{i=10}^{13}\sum_{j=3}^{4}A_{ij}^{-}\cdot U_{ij}^{+}=0$$

As for HPS, $\sum_{i=14}^{15}\sum_{j=1}^{2}A_{ij}^{-}\cdot U_{ij}^{+}\geq \sum_{i=14}^{15}\sum_{j=1}^{2}A_{ij\min}\cdot U_{ij}^{+}$,
(35)$$\sum_{i=14}^{15}\sum_{j=3}^{4}A_{ij}^{-}\cdot U_{ij}^{+}=\sum_{n=3}^{3}Q_{n}^{+}-\sum_{i=14}^{15}\sum_{j=1}^{2}A_{ij}^{-}\cdot U_{ij}^{+}$$
when, $\sum_{i=14}^{15}\sum_{j=1}^{2}A_{ij}^{-}\cdot U_{ij}^{+}\leq \sum_{i=14}^{15}\sum_{j=1}^{2}A_{ij\min}\cdot U_{ij}^{+}$,
(36)$$\sum_{i=14}^{15}\sum_{j=3}^{4}A_{ij}^{-}\cdot U_{ij}^{+}=0$$
(37)$$(W_{mn}^{-}+\Delta W_{mn}z_{mn})\geq S_{mn}^{+}\geq 0,\forall m,n$$
(38)$$S_{mn}^{+}\geq S_{mn}^{-},\forall m,n$$
(39)$$0\leq z_{mn}\leq 1,\forall m,n$$

After solving and calculating $A_{ijopt}^{-}$, $S_{mnopt}^{+}$ and $f_{opt}^{-}$, and combining the two sub-models, the solution of the interval two-stage stochastic programming model is as follows: $A_{ijopt}^{\pm }=[A_{ijopt}^{-},A_{ijopt}^{+}]$, $S_{mnopt}^{\pm }=[S_{mnopt}^{-},S_{mnopt}^{+}]$, $f_{opt}^{\pm }=[f_{opt}^{-},f_{opt}^{+}]$, and the optimal hydration scheme is: $OPT_{mn}^{\pm }=W_{mnopt}^{\pm }-S_{mnopt}^{\pm },\forall m,n$.

### 3.3. Model Parameters and Data Description

In this paper, 15 lakes in the MNNR of Jilin Province were studied, and the planning area of four ecological service functions in the “Planning Report” was used as the first known condition under which the first stage of sufficient water replenishment occurred. It was also assumed that the ecological function structure of the study area was unchanged during the forecast year to determine the water supply target of the first stage. During the calculation of the model, there were three types of flow conditions: low, medium, and high. According to the sequence of water resources in the Nenjiang River Basin of “Jilin Province Water Resources” from 1956 to 2010, we used the Monte Carlo simulation provided by Oracle Crystal Ball (Oracle, Redwood Shores, CA, USA) software to obtain the frequency distribution of the available water resources in the Nenjiang River Basin (Figure 2). It gave the available water resources during the forecast year for the predicted low, medium, and high flow levels, with probabilities of 0.4, 0.5, and 0.1, respectively. We chose 20% (±10%), 65% (±10%), and 95% (±10%) quantile as model input data for the three flow levels (Table 2). Table 3 lists the ecological function planning area, water demand per unit area, and ecological benefits per unit area of different lakes. Table 4 combines the ecological benefits evaluation index system and gives the ecological service types that correspond to different ecological functional areas. The planning area of different lake ecological functions is referred to in Reference [25]. The water demand and ecological benefits per unit area of different ecological functions are referred to in Reference [32,33]. The water transportation cost of different water diversion methods is referred to in Reference [28]. The water shortage penalty coefficient referred to the ecological benefits per unit area (Table 5), and the detailed procedures are provided in Section C in Appendix A.

## 4. Results Analysis and Discussion

### 4.1. Water Supply Allocation Scheme

Based on the interval two-stage stochastic programming method, we constructed an optimal allocation model of flood resources under ecological benefits constraints in this paper and used Lingo11.0 software (Lindo Systems, 1415 North Dayton Street, Chicago, IL, USA) to solve the optimal target of normal water and flood discharge of different water intakes. At the same time, we obtained the water shortage for the forecast year and the optimal allocation under different flow levels. The optimal water supply target, water shortage and optimal water distribution of different water diversion methods under different flow levels are shown in Figure 3.

It can be seen from Figure 3 that for the same type of water resources of the same water intake, the optimal water supply target is the same under different flow levels. Normal water diversion and flood diversion targets of TBC are 19.21 and 31.88 million m^3^, respectively. The type of water resources of the SIG and HPS water intake are only flood, and the flood diversion targets are 24.33 and 4.04 million m^3^. For the TBC water intake, when the forecast year is high flow level, the optimal allocation of water is normal water diversion the range of 17.39 to 18.16 million m^3^, and flood diversion the range of 29.40 to 30.97 million m^3^; when the forecast year is medium flow level, the optimal allocation of water is normal water diversion the range of 16.33 to 17.37 million m^3^, and flood diversion the range of 28.55 to 30.63 million m^3^; when the forecast year is low flow level, the optimal allocation of water is normal water diversion the range of 15.35 to 16.33 million m^3^ and flood diversion the range of 27.62 to 28.83 million m^3^. The optimal allocation of water is determined by the difference between the optimal water supply target and the water shortage. To reduce the water supply cost and save resources under different flow levels, the model makes a choice of water shortage. Therefore, in the implementation process of the West Water Supply Project of Jilin Province, controlling the operating cost of the project will help reduce the water shortage of the model and achieve greater ecological benefits. In addition, for the same type of water resources of the same water intake, as the water volume increases, the water shortage will gradually decrease, the water supply in the MNNR will increase continuously, and the optimal water distribution will gradually increase.

The decision variables of TBC are *z_11opt_* = 0.74, *z_12opt_* = 0.81, and the decision variables of SIG and HPS are *z_22opt_* = 0.85, *z_32opt_* = 0.82. The decision variables are all close to 1, which indicates that the optimal water supply targets are close to the upper limit of the initial water distribution targets. It shows that for the three water intakes, the economic benefits from water diversion to ensure the ecological function of the lake is higher than the water supply cost due to the increase of the water diversion. This means that the model chose to prioritize the various ecological functions of different lakes to give full support to the ecological benefits of the MNNR. In the case of uncertain water supply, the risk of meeting high water demand is large, and the penalty when water consumption has not been met is also large, while the low water demand is reversed. However, the economic benefits that are generated by low water demand will be less, which indicates that water supply, risk, and economic benefits are closely linked.

In addition, for the TBC water intake, *z_11opt_* < *z_12opt_* indicates that flood diversion is preferred in the process of water diversion. This is because the flood cost is lower than the cost of normal water resources and the utilization of flood resources can produce greater economic benefits. The optimization results show that the amount of flooding accounts for 77% of the total water diversion, which has increased by 24% compared with the water diversion scheme in the “Planning Report”. The difference in the amount of normal water to that of the water produced by floods of different water intakes before and after optimization is shown in Figure 4.

It can be seen from Figure 4 that the optimal allocation of water in the MNNR is lower than that of the planned water volume. At low, medium, and high flow levels, the total amount of water diversion after optimization decreased by the range of 7.03 to 12.12 million m^3^, the range of 36.74 to 44.64 million m^3^, and the range of 45.12to 54.92 million m^3^, respectively. For the three water intakes, the decrease of TBC is the most significant, especially for normal water diversion; TBC is reduced by the range of 56% to 57%, the range of 68% to 69%, and the range of 69% to 70% at different flow levels. However, the amount of flood diversion of TBC increased by the range of 3.84 to 5.01 million m^3^ at low flow levels, because the ecological function water demand of the lakes is met at a low flow level, and considering the ecological benefits constraint, the model selected the lower cost resources to replace part of the normal water resources and to maximize water supply benefits. For the SIG water intake, the amount of flood diversion increased significantly at low flow levels, while the change was not significant under the medium and high flow levels. For the HPS water intake, the amount of flood diversion increased under the three flow levels, which may be because the lakes corresponding to HPS are Nashitu and Baoshan lakes, which are located on the edge of the MNNR, adjacent to the Nenjiang River, and the reeds in the lakes are widely distributed. Due to the high biomass per unit area of reed wetland, the water demand of reed wetland is higher than that of fish pond, crab pond and marsh ecological function areas, thus resulting in the HPS optimal allocation of water being higher than that of the planned water volume.

### 4.2. Ecological Benefits Analysis of the Water Replenishment Scheme in Momoge National Nature Reserve (MNNR)

According to the survey data of the environmental status of the MNNR that is provided in the “Environmental Impact Report of the West Water Supply Project of Jilin Province”, combined with the planned area of the ecological function area in the “Planning Report” and the ecological benefits evaluation index system established in this paper, the interval two-stage stochastic programming model was used to solve the distribution of normal water and flood diversion under different flow levels, and the ecological benefits arising from the implementation of the western water supply project under the corresponding conditions were obtained.

The model calculation results show that at low flow levels, the annual water intake of TBC is the range of 42.97 to 45.16 million m^3^, the water intake of SIG is the range of 22.32 to 22.78 million m^3^, and the annual water intake of HPS is the range of 3.22 to 3.44 million m^3^. Figure 5 shows the lower and upper limits of the area of different ecological functional areas of 15 lakes at low flow levels.

The shaded parts in Figure 5 are the upper and lower limits of the planned area of each of the 15 lakes in the MNNR in the “Planning Report”. The blank parts are the added values of the results of the model optimization relative to the planned area values, and the digital labels are the added area values. It can be found that under the low flow level, the model-optimized water supplement schemes did not significantly increase the area of the lakes’ ecological functional zones, which were roughly the same as the planned area. This was mainly due to the large amount of water shortage at low flow levels and the high cost of replenishment. To meet the ecological benefits, it was necessary to ensure the maximum economic benefits of the western water replenishment project simultaneously. Therefore, the model chose to meet the planned area value, but did not increase the amount of water to obtain greater ecological benefits. According to the evaluation index system of the ecological service function constructed in this paper, the ecological benefits of the MNNR can be obtained with the range of 26.30 to 32.14 million yuan under the low flow level.

Under the middle flow level, the water intake at the three intakes increased by approximately 2–6%. Figure 6 shows the different ecological functional areas of 15 lakes at medium flow levels.

It can be found from Figure 6 that, compared with the four ecological functions of the lake bubble at the low flow level, those functions increased at the middle flow level. This shows that at a higher flow level, the additional water cost was reduced due to the decrease in water shortage, and the model selected higher ecological benefits, which maximized the economic benefits of the West Water Supply Project of Jilin Province. In addition, compared with different ecological functional areas, it can be seen that the increase in the area of fish ponds was the lowest, which was mainly because only Haernao Lake among the 15 lakes has the function of raising fish. By comparing the crab pond, reed wetland, and marsh wetland, it can be found that under the medium flow level, the crab pond area, the reed wetland area, and the marsh wetland areas increased by the range of 0.57 to 0.70 million m^2^, the range of 0.95 to 1.16 million m^2^, and the range of 1.83 to 2.24 million m^2^, respectively. The increase in the marsh wetland area was the largest, the increase of the reed wetland area was the second largest, and the increase of the crab pond area was the smallest. This is mainly due to the ecological functions of vegetation carbon sequestration and oxygen release in marshes and reed wetlands. The two eco-service functions have a higher unit area income, and the area of each ecological function area of the MNNR cannot be expanded indefinitely. Therefore, the model allocates a limited area to the ecological function area with higher economic benefits; that is, the area of the marsh wetland > the area of reed wetland > the area of crab pond > the area of fish pond. In summary, the ecological benefits of the MNNR is the range of 28.21 to 33.49 million yuan under the medium flow level.

As in the middle flow level, the same rule is reflected in the high flow level, as shown in Figure 7; the increase in the reed wetland and marsh wetland area is higher than the increase of the crab pond and fish pond areas. By comparing the different lakes, it can be found that the areas of Yuanbaotu Lake, Etou Lake, Gaomian Lake, and Haernao Lake had larger added values, among which Haernao Lake had the largest increase, and the total area of the four ecological functional areas increased by the range of 1.78 to 2.17 million m^2^. As a result, the ecological benefits of the MNNR is the range of 29.41 to 35.94 million yuan under the high flow level.

## 5. Conclusions

Based on the maximum ecological benefits, we constructed a flood resource optimal allocation model based on the interval two-stage stochastic programming method in this paper. The distribution plan of normal water diversion and flood discharge in the West Water Supply Project of Jilin Province under different flow levels was obtained, and the ecological benefits under the corresponding situation were given.

The optimal allocation of water in the MNNR decreased significantly compared with the total planned water volume, and as the flow level increased, the total optimal water volume increased compared with the planned water volume. At low, medium, and high flow levels, the total amount of water diversion after optimization decreased by the range of 7.03 to 12.12 million m^3^, the range of 36.74 to 44.64 million m^3^ and the range of 45.12 to 54.92 million m^3^, respectively. In terms of the utilization of flood resources, the proportion of flood resources that is cited in the MNNR in the different scenarios is more than 70%. In terms of the utilization of flood resources, the proportion of flood resources in the MNNR are more than 70% after optimization at different flow levels. Among them, the proportion of flood resources at a low flow level reached 77%, which was 23% higher than that under the plan, which indicated that when the forecast year is low flow level and the amount of incoming water is small, the utilization of flood resources is crucial for the ecological compensation of protected areas.

In addition, the ecological benefits of the West Water Supply Project of Jilin Province at low, medium, and high flow levels reached the range of 26.30 to 32.14 million yuan, the range of 28.21 33.49 million yuan, and the range of 29.41 to 35.94 million yuan, respectively. According to the model optimization results, flood resources can significantly reduce the utilization of normal water resources and meet the water requirements of fish, crabs, polder wetlands, and marsh wetlands in the MNNR, which can be an effective supplement to water resources. The model optimization results can also provide a basis for the allocation of transit flood resources in other regions.

In this paper, we used the interval two-stage stochastic programming method to allocate normal water and flood resources under three flow levels. However, this model did not consider the uncertainty of incoming water quality and made some assumptions in the calculation of water diversion cost and economic benefits, which inevitably brought errors to the system. Further research is needed to solve the ecological compensation problem of flood resources.

## Figures and Tables

**Figure 1 ijerph-16-01033-f001:**
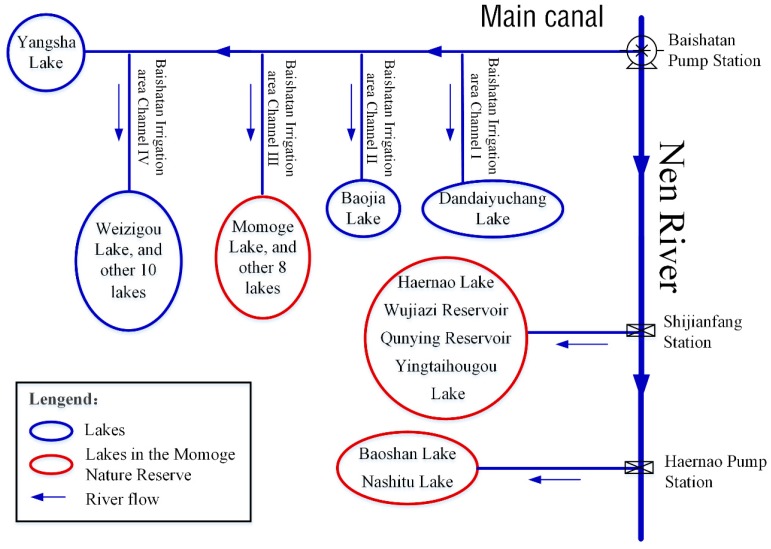
Schematic diagram of Nenjiang water supply node in the Momoge National Nature Reserve (MNNR).

**Figure 2 ijerph-16-01033-f002:**
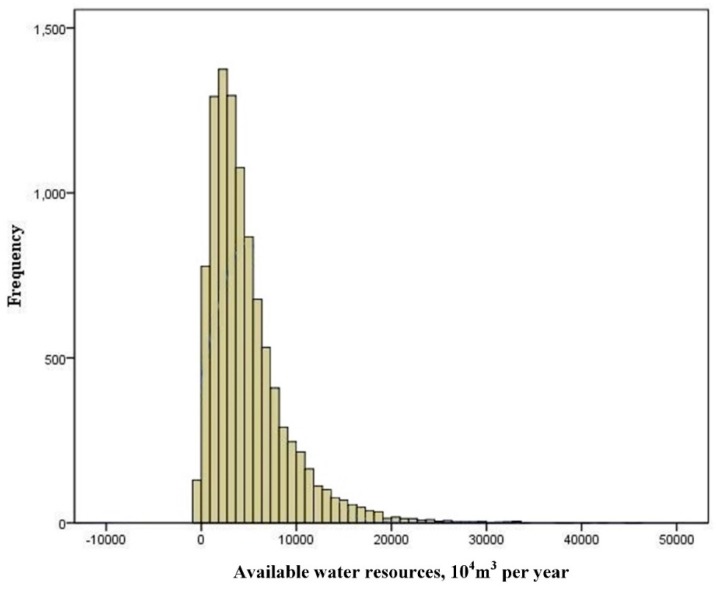
Frequency distribution of available water resources in the Nenjiang River Basin.

**Figure 3 ijerph-16-01033-f003:**
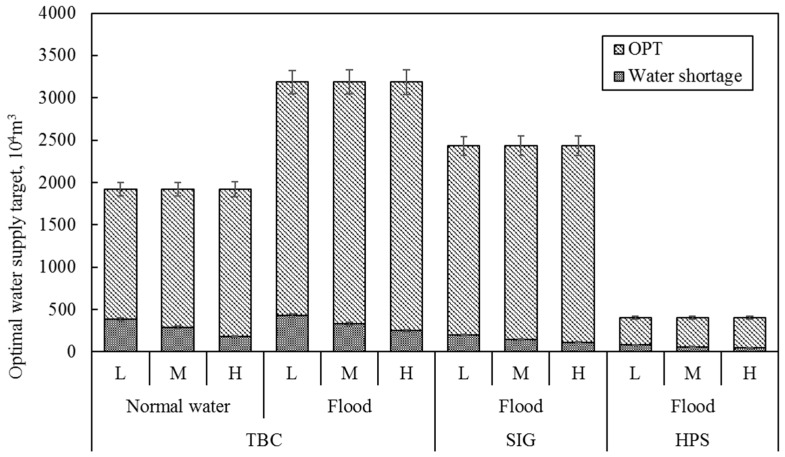
Water supply by different water diversion methods at each water intake of the Momoge National Nature Reserve (MNNR) in different scenarios.

**Figure 4 ijerph-16-01033-f004:**
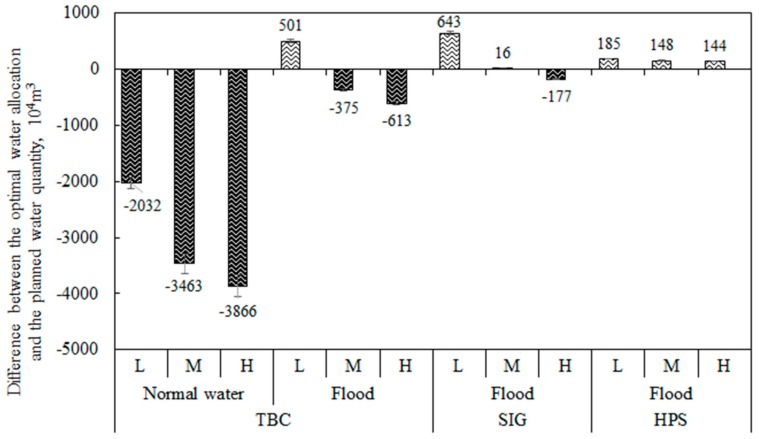
The reduction of the optimal water allocation compared with the planned water quantity of different water diversion resources in the water intakes of the Momoge National Nature Reserve (MNNR) under different flow levels (Negative values indicate decrease, and positive values indicate increase).

**Figure 5 ijerph-16-01033-f005:**
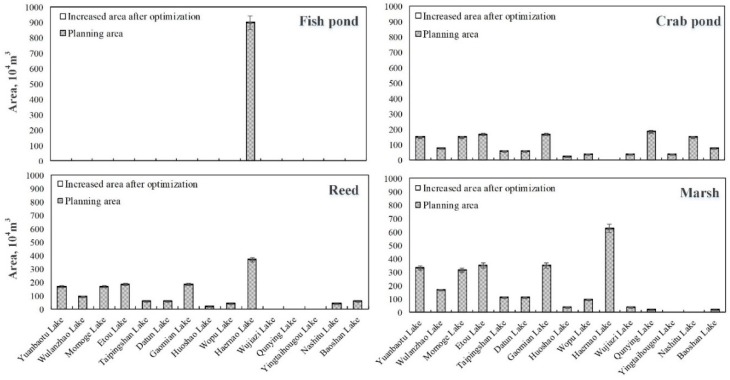
The area of each ecological functional area of the lakes at a low flow level (the shaded part is the planned value; the blank part is the added value after optimization).

**Figure 6 ijerph-16-01033-f006:**
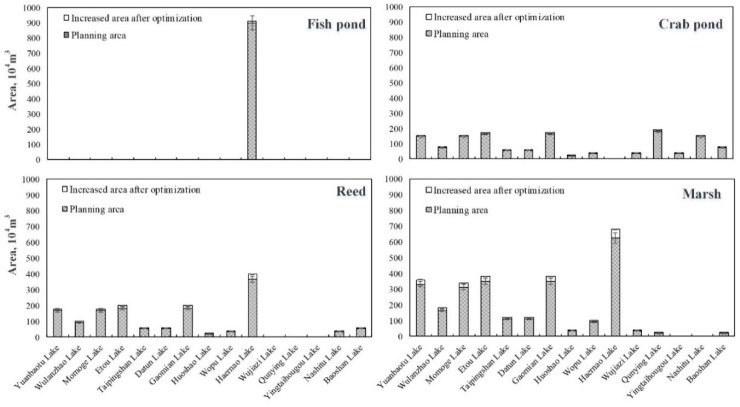
The area of each ecological functional area of the lakes at medium flow level (the shaded part is the planned value; the blank part is the added value after optimization).

**Figure 7 ijerph-16-01033-f007:**
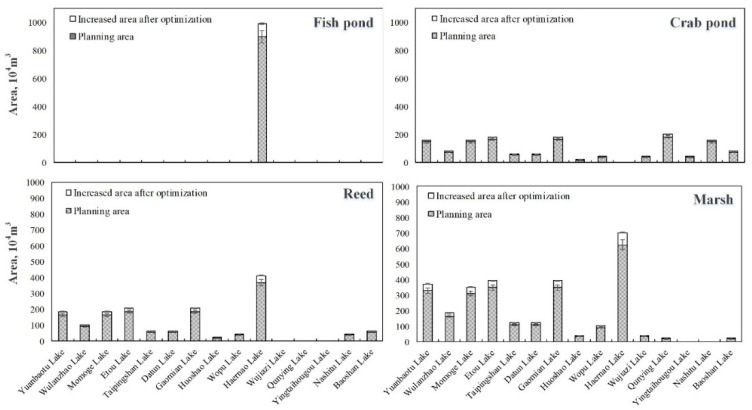
The area of each ecological functional area of the lakes at high flow level (the shaded part is the planned value; the blank part is the added value after optimization).

**Table 1 ijerph-16-01033-t001:** Wetland ecological service function evaluation index system.

Target Layer	Criteria Layers	Indicator Layers
Total benefits of ecological water transfer project	Ecosystem material production	Food production value
		Raw material production value
	Ecosystem environmental regulation	Carbon sequestration value
		Oxygen release value
		Flood storage value
		Microclimate regulation value
		Plant adsorption value
		Biodiversity value
	Humanities and social services	Scientific research value
		Tourism value
		Landscape value

**Table 2 ijerph-16-01033-t002:** Water intake of each water intake at different flow levels in Nenjiang River.

Water Intake	Water Flow Level	Normal Water (10^4^ m^3^ per Year)		Flood (10^4^ m^3^ per Year)	
		Lower Bound	Upper Bound	Lower Bound	Upper Bound
TBC	*L* (*P_1_* = 0.4)	2060	4634	1306	2938
	*M* (*P_2_* = 0.5)	4291	6437	2720	4080
	*H* (*P_3_* = 0.1)	5149	10621	3264	7344
SIG	*L* (*P_1_* = 0.4)	0	0	918	2065
	*M* (*P_2_* = 0.5)	0	0	1912	2868
	*H* (*P_3_* = 0.1)	0	0	2294	5162
HPS	*L* (*P_1_* = 0.4)	0	0	79	178
	*M* (*P_2_* = 0.5)	0	0	165	247
	*H* (*P_3_* = 0.1)	0	0	198	445

Note: TBC, SIG and HPS represent the third branch channel of Baishatan irrigation area, Shijianfang intake gate and the Haernao pumping station, respectively. L, M and H represent the low, medium and high water flow level.

**Table 3 ijerph-16-01033-t003:** Ecological function planning area, water demand and ecological benefits per unit area.

	Water intake	Lakes	Fish Pond		Crab Pond		Reed Wetland		Marsh	
			Lower Bound	Upper Bound	Lower Bound	Upper Bound	Lower Bound	Upper Bound	Lower Bound	Upper Bound
Ecological function planning area (10^4^ m^2^)	TBC	Yuanbaotu Lake	-	-	120	147	135	165	270	330
		Wulanzhao Lake	-	-	60	73	75	92	135	165
		Momoge Lake	-	-	120	147	135	165	255	312
		Etou Lake	-	-	135	165	150	183	285	348
		Taipingshan Lake	-	-	45	55	45	55	90	110
		Datun Lake	-	-	45	55	45	55	90	110
		Gaomian Lake	-	-	135	165	150	183	285	348
		Huoshao Lake	-	-	15	18	15	18	30	37
		Wopu Lake	-	-	30	37	30	37	75	92
	SIG	Haernao Lake	735	898	-	-	300	367	510	623
		Wujiazi Lake	-	-	30	37	-	-	30	37
		Qunying Lake	-	-	150	183	-	-	15	18
		Yingtaihougou Lake	-	-	30	37	-	-	-	-
	HPS	Nashitu Lake	-	-	120	147	30	37	-	
		Baoshan Lake	-	-	60	73	45	55	15	18
Water demand per unit area (m^3^/m^2^)			0.5	0.6	0.5	0.6	1.7	2.9	0.7	1.1
Ecological benefits per unit area (Yuan/m^2^)			0.04	2.40	0.04	3.60	0.04	3.64	0.04	3.64

**Table 4 ijerph-16-01033-t004:** Types of ecological services corresponding to different ecological functions in nature reserves.

	Fish Pond	Crab Pond	Reed	Marsh
Food production (Fish)	1	0	0	0
Food production (Crab)	0	1	0	0
Raw material production (Reed)	0	0	1	0
Carbon sequestration	0	0	1	1
Oxygen release	0	0	1	1
Flood storage	1	1	1	1
Microclimate regulation	1	1	1	1
Plant adsorption	0	0	1	1
Biodiversity	1	1	1	1
Scientific research	1	1	1	1
Tourism	1	1	1	1
Landscape	1	1	1	1

Note: “1” indicates that the ecological function area has the ecological service, and “0” means there is no such service.

**Table 5 ijerph-16-01033-t005:** Water diversion cost [28] and water shortage penalty coefficient of different water diversion methods (Section C in Appendix A).

Water Intake	Type of Water Resources	Cost (Yuan/m^3^)	Water Shortage Penalty Coefficient (Yuan/m^3^)	
			Lower Bound	Upper Bound
TBC	Normal water	0.18	0.04	3.64
	Flood	0.11	0.04	3.64
SIG	Normal water	0.18	0	0
	Flood	0.11	0.04	3.64
HPS	Normal water	0.18	0	0
	Flood	0.11	0.04	3.64

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
