# Peer review of "An Interval Two-Stage Stochastic Programming Model for Flood Resources Allocation under Ecological Benefits as a Constraint Combined with Ecological Compensation Concept"

_ijerph, 2019, doi:10.3390/ijerph16061033_

Round 1

Reviewer 1 Report

Summary

    The authors present a work that looks at an optimal water resource allocation method during normal and flooded conditions with a focus on costs associated with water diversion and water shortages to ecosystem services. Four ecosystems are examined, each with its own set of services provided. The optimization takes into account the benefit of maintaining healthy ecosystem services by weighing it against the cost of moving water (diversion cost) and the cost of not moving enough water (shortage cost; stress on environment). No other costs or benefits appear to be taken into account. Of particular interest is the lack of water quality, which should have an impact on the ecological benefits provided, especially in a work that is based on restorative processes for ecosystems.

    Due to uncertainties in some of the model parameters, the authors utilize a two-stage interval approach that provides upper and lower bounds for their model solutions. As a part of this work, the authors calculate the probabilities associated with high, normal, and low flow conditions of their study river (reasoning for these three defining levels is not entirely clear), which are based on Monte Carlo simulations (methods not described; however, this is likely to account for the highly managed water resources upstream of their study area). 

    The water allocation in this study is designed for several lakes located within a nature reserve along the Nenjiang River in China. The existing diversion network of canals and pumping stations are shown in Figure 1 (labels and explanations are incomplete). The model description introduces several undefined concepts, such as "ecological function structure," "water resource carry capacity," "water source supply target value," and "water replenishment benefit," which do not appear in the model equations; otherwise, the model formulation appears logical and was easy to follow.

    The significant values used to run the model are presented in Tables 2 through 5. The valuation methods used for populating these tables (and any uncertainties or limitations given thereto) are prescribed to the so called "Planning Report" (no citation given). Trusting these values, the model is run for a characteristic year, which includes three flow conditions under both the normal and flood season.

    Optimal allocation results are given for the three main diversion channels that lead to the lakes within the nature reserve. Results are presented mainly as bar graphs, which emphasize the differences among the three diversion intakes for the three flow conditions, but are not as clear at showing the reader a comparison of the upper and lower limits (the uncertainty) of the interval model results. The authors compare their optimization results against those in the "Planning Report," which includes an examination of the change in pumping rates at the three diversion intakes and the different prescribed areas for the four ecosystems. The authors show marginal positive impacts based on their optimization scheme; however, actual benefits from this proposed scheme are uncertain. 

Major issues

    • The authors compare their optimization method against what was presented in the "Planning Report." How were the values in the "Planning Report" determined? It is difficult to compare two outcomes from only one known algorithm.

    • The novelty of the model presented by the authors is the introduction of the water shortage penalty coefficient; however, it is not clear how this coefficient is quantified. The paper indicates that it is based on the ecological benefit per unit area (see Table 3), but are they weighted values? If so, how are the weights calculated (the number of lakes serviced by the diversion intake, the lake areas, the ecosystems they have planned)? 

    • The authors make a special point to using the interval modeling technique that captures uncertainty in model parameters; however the present use of bar graphs does not do a good job of showing the uncertainty in the results. Is there a better way to show the upper and lower bounds of your results? One possible way would be to use something similar to an average +/− the standard deviation.

    • There is an obvious lack of methodology presented in this work (particularly in Section 3.3), arguably to the extent that I do not believe the results could be reproduced by an independent party. The authors are strongly encouraged to include where and how values were derived. 

    • For a paper that is being submitted to an open access journal, over 70% of the references are either unsearchable (could not be located online) or are behind a paywall. I urge authors to include DOIs or URIs where applicable (I found a DOI/URI for 12 of the references) and to consider the accessibility of their literature sources.

    • To show significance of this work, the authors mention that most studies are based on a feasibility analysis and that few are based on practice (see Lines 62-64). My question is how is this study any different? What is the significance of this work?

    • Will the authors be releasing their Lingo11 code?

Minor concerns

    • In the Introduction, what is meant by "ecological compensation" and what is an "ecological compensation system"?

    • It is indicated in the model constraints that the ecological service functions must be limited to the lake and cannot be expanded indefinitely. My question is how large are each of the fifteen lakes in the study? It may be more interesting to see how proportions of each lake are being utilized rather than the raw numbers.

    • Lines 52-54: The sentence that begins "The use of flood resources in the basin is an important measure to exploit the characteristics of..." is unclear in its meaning. What are you exploiting? 

    • Lines 66-69: From the sentence that begins "Studying the allocation of flood resources in various fields...," what industries are you referring to?

    • Lines 70-90: It is not clear how any of these studies are related to flood water resources or their utilization. It is strongly encouraged that the authors introduce the significance of these studies and how they relate to this work.

    • Lines 100-112: In this paragraph, water diversion is given priority over what? I'm not certain I understand why, under flood conditions, water diversion is necessary to ensure the water needs of the Momoge National Nature Reserve.

    • Figure 1: What do the numbers indicate for the pump stations and irrigation channels and do they have units? Could you explain what is meant by "Flood" and "Normal" along the main canal and why there are four sets of values given? The numbering (I, II, III, IV) for the four Baishatan Irrigation Area Channels are sideways compared to the textual labels, which makes them difficult to read/understand. Please consider orienting them to be inline with the rest of the text.

    • Somewhere in Section 2.1, it may be useful for the reader to know that your study area is located in wetlands (ecosystem type).

    • In Section 2.2, the authors may find some additional useful references for defining and relating ecosystem services, for example:

        ◦ Joseph Alcamo et al. (2003). Ecosystems and human well-being: a framework for assessment. Island Press, Washington D.C. https://www.cifor.org/library/1866/ 

        ◦ Raffaelli and White (2013). http://dx.doi.org/10.1016/B978-0-12-417199-2.00001-X 

    • There is nothing in Section 2.2 on what is the evaluation index or how it is measured.

    • The two paragraphs that begin Section 3 are difficult to understand due to the introduction of several concepts in a somewhat disorganized manner. Consider simplifying the ideas and defining terms and/or giving examples. Such as "the optimal allocation of water resources in a complex problem due to several contributing factors with inherent randomness (for example…)." It is unclear what "ecological function structure," "water resource carry capacity," "water source supply target value," "water replenishment benefit" are pertaining to.  

    • Throughout this work, the authors use several variants to describe the Momoge National Nature Reserve (e.g., "Water Nature Reserve," "Momoge Conservation Area," "Momoge protection zone," "Momoge Reserve," "Momoge Nature Reserve," and "Momoge National Nature Reserve"). To avoid confusion, consider using an abbreviation, for example "MNNR" or "Momoge Reserve," and use it consistently.

    • In Section 3.1, the authors have two definitions for the term T: water transfer cost (Line 170) and water supply cost (Line 187). For clarity, consider using one definition.

    • Line 170: This is the first mention of fish and crab production and it is a little confusing as a reader how it relates to the uncertainty of the water deficit penalty coefficient. Consider removing them from this list or explain their significance. 

    • It is unclear what the reasoning is behind making H = 3. Could the authors elaborate on this decision?

    • Lines 187-188: Are the units for T and C supposed to be yuan per cubic meter?

    • It is unclear how the ranges for water demand (U) and ecological benefit (B) presented in Table 3 are utilized in the model; these variables appear to only be dependent upon the specific lake and ecological service function. Could the authors please elaborate on how these ranges are used?

    • Lines 299-300: Can the authors provide any reasoning why the water supply target does not change with changing flow conditions?

    • Lines 340-342: It is unclear whether Figure 4 shows that the optimal allocation of water is significantly lower than planned; how is significance being defined (statistical)? Furthermore, it is unclear that Figure 4 shows that as flow level increases (i.e., from L to M to H), the optimal allocation of water increases compared to the planned water volume. If this were the case, in my impression, the bar graphs should become larger in magnitude (difference equals optimal minus planned); however, in each case, the magnitude of the bars decreases, which indicates to me that either optimal is decreasing and/or planned water volume is increasing.  Could the authors please elaborate on this further? It is encouraged that the authors also change the label for the y axis to reflect that these values are differences.

    • Lines 424-426: Could the authors please elaborate on how the ecological benefit of migratory birds comes in to play with the ecological benefit model?

Minor edits

There are some issues with the English grammar, especially later in the manuscript. Some corrections are listed below for the authors' consideration.

    • Line 72. Consider "of different water qualities."

    • Lines 100-101. For clarity, consider the following revision: "The West Water Supply Project of Jilin Province is located in the western plain area on the right bank of Nenjiang River and on the left bank of Songhua River and covers 8 counties."

    • Line 110. Typo. Consider changing "There are three water intakes for the Momoge Nature Reserve and..." to "The three water intakes for the Momoge Nature Reserve are..."

    • Line 120. Typo. Colons precede a list. Consider changing "Ecosystem service functions can be summarized into three categories: the first is the production of substances in the ecosystem, including the functions that the wetland ecosystem can provide for human beings, such as food, industrial raw materials, etc." to "Ecosystem service functions can be summarized into three categories. The first category is the production of substances in the ecosystem, including the functions that the wetland ecosystem can provide for human beings, such as food, industrial raw materials, etc."

    • Lines 131-132. Consider moving the sentence "The West Water Supply Project of Jilin Province is an ecological water transfer project..." to Section 2.1 where the project is introduced.

    • Line 171. Consider using "water shortage penalty coefficient" to be consistent with the rest of this paper.

    • Equation 7: missing the starting value of j in the summation

    • Equation 12: the right-alignment of the equation number is off

    • Line 213. Consider a period rather than a colon at the end of the sentence.

    • Line 226-227. Consider separating this sentence at the end of Line 226 a period and starting Line 227 as a new sentence.

    • Lines 243. Please check that you meant "upper limit" here.

    • Line 258. Awkward wording "taken as research objects." 

    • Line 258. Please confirm that you meant 15 ecological functional zones; do you mean four ecological service functions?

    • Lines 274-277. Missing reference to Table 5.

    • Line 276. Inconsistency with italics for "Planning Report"

    • Line 279. Consider rephrasing the figure caption from "Probability distribution" to "Frequency distribution" or "Histogram." Also, when you present more than one dataset in a figure, please consider defining each in the caption or presenting them in a legend.

    • Line 280. Consider including in the table caption the definitions for the intake and flow level abbreviations and note why a range of values are given for each (are these the mean +/- standard deviation from the Monte Carlo simulations?). Also, assuming the abbreviations of H (high), M (medium), and L (low), there is an inconsistency with labeling the flow levels to their probability levels. In the text (Line 268), it says high, medium, low levels of 0.4, 0.5, and 0.1, respectively. In Table 2, it shows L (P1=0.4), M (P2=0.5), and H (P3=0.1)---in reverse order. 

    • Line 290. Consider including the software developer along with the software name.

    • Line 333. Typo. Consider "produced by floods" rather than "produces by floods"

    • Line 337 (Figure 4 caption). It may be helpful to the reader to note that the difference between optimal allocation and planned water quantity is defined as water shortage (S) based on the equation on Line 255.

Author Response

Thank you very much for your comments and suggestions on our paper submitted to “International Journal of Environmental Research and Public Health”. We have revised the manuscript entitled “An Interval Two-stage Stochastic Programming Model for Flood Resources Allocation under Ecological Benefits as a Constraint Combined with Ecological Compensation Concept” (ID: ijerph-447635), and would like to re-submit it for your consideration. We have checked the manuscript and revised it according to the comments. We hope that the revision is acceptable, and I look forward to hearing from you soon.

Reviewer 2 Report

The study is generally well written and seems to be scientifically sound. The language is good in most parts, but in the introduction and in the methodology sections there are some confusing parts, or parts with complicated syntax that are difficult to follow (e.g. too long and complicated sentences). A characteristic example is the presentation of the main objectives of this study in Lines 91-97.

The problem that is solved is well stated and the followed analysis and model construction seem to be rational.

On the other hand, the paper mostly presents a case study. As can be seen in the results, while the described results justify the usefulness of the applied methodology, the presentation of the results mostly focus on the specific quantities, costs, water allocations etc. and not to the effectiveness of the method, its characteristics pros and cons etc. The results are reasonable but there are no comparisons with alternative possible methodologies – solutions, no information about novel approaches or features and the final conclusions mostly refer to the case study than to the methodology pros and cons, characteristics, novelties etc.

Finally, I believe that the topic of the manuscript is not very well suited to this journal, at least based on the aims and scope of the journal; however, the editors are more capable to decide this.

Conclusively, this study is well written and mostly interesting (though for an international reader the interest of the long descriptions of the results and the conclusions is limited). However, it is mostly a case study; therefore, I suggest that in case that this journal publishes case studies and the topic of this paper is considered to lay in the scope of the journal, then it may be published after a minor revision (mostly clarity, linguistic improvements). Otherwise I believe that it should be rejected and submitted to a more appropriate journal.

Author Response

(The authors gave the same response as above.)

Round 2

Reviewer 1 Report

I would like to thank the authors for their time and efforts addressing each of the concerns that were raised. Their responses are satisfactory. Overall, I find the manuscript to be much improved. I feel that following clarification of a few minor issues listed below that this article is worthy of publication.

Line 107: consider using the MNNR abbreviation here

Line 130: authors mention 12 service function evaluation index systems, but list only eight and, in Table 1 and Table S1, list only 11. Based on Table 4, it would seem the twelfth service function index system is based on the separation of crab and fish food production. Would the authors please confirm these numbers are correct and make necessary clarifications?

Line 257–258: authors may want to point out that the values, 0.1, 0.5, and 0.4 correspond to probabilities.

Lines 263–266: authors are still missing a reference to their Table 5.

Figure 5: authors may wish to indicate in the table caption that the y-axis scales are all different; the currently mismatched scales may mislead readers, for example, under brief review, it looks like the area of reeds is the same as the area of marshes for Haernao Lake; however, the area of reeds is actually about 40% less than that indicated for marshes, despite the bars appearing the same size. To reduce further confusion, it is also recommended that the y-scale not change between Figures 5, 6 and 7 (e.g. 1000 to 1200 for fish pond and 700 to 800 for marsh).

Line 410: it is a little confusing to use the word "genetic" to describe your algorithm, as it is not used anywhere else in your paper.

In Table S1, it may be beneficial to readers to include the citations for the evaluation methods as a quick reference.

In Supplemental Material Section C, can the authors clarify what is referenced as "Question 1"?

Also in this section, following Table S2, I would recommend removing the reference to weights as it may confuse readers why you are mentioning this.

p { margin-bottom: 0.1in; line-height: 120%; }a:link { }

Author Response

Dear Reviewers:

Thank you very much for your comments and suggestions on our manuscript submitted to “International Journal of Environmental Research and Public Health”. Those comments are valuable and helpful for revising and improving our manuscript, as well as the important guiding significance to our researches. We have revised the manuscript entitled “An Interval Two-stage Stochastic Programming Model for Flood Resources Allocation under Ecological Benefits as a Constraint Combined with Ecological Compensation Concept” (ID: ijerph-447635), and would like to re-submit it for your consideration. We hope that the revision is acceptable, and look forward to hearing from you soon.

Thank you and best regards.

Yours sincerely,

Yu Li

Affiliation: College of Environmental Science and Engineering, North China Electric Power University, Beijing

Reviewer 2 Report

As I wrote in my previous review, the study is well written and seems to be scientifically sound. The language is good and some additional improvements have been done in the revised version. The authors also replied to all the comments of the other reviewer and made the required  revisions (however limited) to address these comments.

The problem is also well stated and the followed analysis and model construction seem to be rational.

My main hesitations, namely that the paper mostly presents a case study and has limited novelty and that it is possibly not very well suited in this journal based on the aims and scope of the journal, are still remaining.

However, the fact that the editors decided to give the opportunity to the authors to submit a revised version indicates that the editors consider that the fact that this study is mostly a case study not to be a problem and they also consider that the manuscript lays in the scope of the journal.

Accordingly, I believe that despite my reservations, it can be published in its present form.

Author Response

Dear Reviewers:

Thank you very much for your generous suggestions entitled “An Interval Two-stage Stochastic Programming Model for Flood Resources Allocation under Ecological Benefits as a Constraint Combined with Ecological Compensation Concept” (ID: ijerph-447635). Those comments are valuable and helpful for revising and improving our manuscript, as well as the important guiding significance to our researches.

Thank you and best regards.

Yours sincerely,

Yu Li

Affiliation: College of Environmental Science and Engineering, North China Electric Power University, Beijing